

# Using Vertical Phase Differences to Better Resolve 3D Gravity Wave Structure

Corwin J Wright[1], Neil P Hindley[1], M Joan Alexander[2], Laura A Holt[2], and Lars Hoffman[3]

[1]Centre for Space, Atmospheric and Oceanic Science, University of Bath, UK
[2]Northwest Research Associates, Colorado, USA
[3]Jülich Supercomputing Centre, Forschungszentrum Jülich, Jülich, Germany

**Correspondence:** Corwin Wright (c.wright@bath.ac.uk)

**Abstract.** Atmospheric gravity waves (GWs) are a critically-important dynamical process in the terrestrial atmosphere, with significant effects on weather and climate. They are geographically ubiquitous in the middle and upper atmosphere, and thus satellite observations are key to characterising their properties and spatial distribution. Nadir-viewing satellite instruments characterise the short-horizontal-wavelength portion of the GW spectrum, which is important for momentum transport, well; however, these nadir-sensing instruments have coarse vertical resolutions. This restricts our ability to characterise the 3D structure of these waves accurately, with important implications for our quantitative understanding of how these waves travel and how they drive the atmospheric circulation when they break. Here, we describe, implement and test a new spectral analysis method to address this problem. This method is optimised for the characterisation of waves in any three-dimensional dataset where one dimension is of coarse resolution relative to variations in the wave field, a description which applies to GW-sensing nadir-sounding satellite instruments but which is also applicable in other areas of science. We show that this "2D+1 ST" method provides significant benefits relative to existing spectrally-isotropic methods for characterising such waves. In particular, it is much more able to detect regional and height variations in observed vertical wavelength, and able to properly characterise extremely vertically long waves that extend beyond the data volume.

## 1 Introduction

The Earth's atmosphere is extremely shallow in the vertical relative to the horizontal, with a depth of tens of kilometres for the well-mixed portion compared to a planetary circumference of 40 000 km. However, the strong effect of gravity causes atmospheric structure and dynamics to vary dramatically over this comparatively short vertical range. For example, the atmosphere has an e-folding scale height distance of only 7 km, while gravity waves with vertical wavelengths ten times that depth transport significant momentum fluxes to the upper atmosphere.

This challenge becomes particularly acute when using stratospheric data from nadir-viewing infrared-sensing satellite instruments such as the Atmospheric Infrared Sounder (AIRS) on NASA's Aqua satellite and the Infrared Atmospheric Sounding Interferometer (IASI) instruments on ESA's MetOp satellites (Aumann et al., 2003; Blumstein et al., 2004; Chahine et al., 2006; Hilton et al., 2012; Hoffmann et al., 2014). These instruments are optimised for measuring horizontal structures with scales of tens of km or less, but typically have vertical resolutions within the same order of magnitude, i.e. several km or





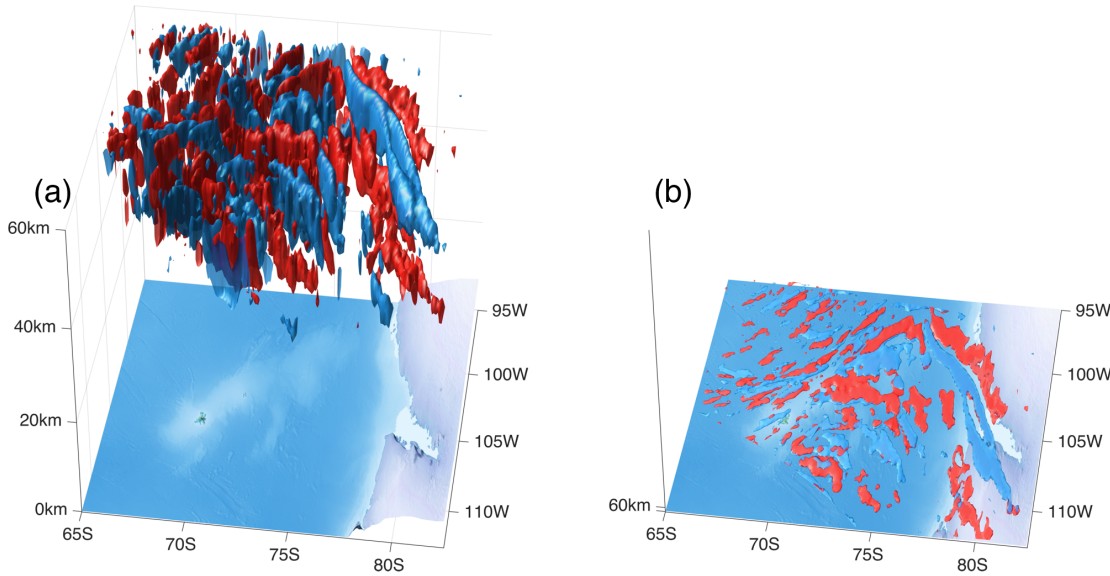

**Figure 1.** 3D sketches of a gravity wave observed on the 28th of June 2010 over the Southern Ocean, shown (a) with the vertical axis stretched to highlight wave structure and (b) to true vertical scale. Red/blue isosurfaces enclose the +/-3.5 K 3D contours respectively. Viewing angle and plotted isosurface magnitudes are identical for both panels.

greater. Since the instruments typically have swath widths of over 1000 km, while information on gravity waves is only useful in the stratosphere between altitudes ∼20–55 km (i.e. a depth of only ∼35 km), the data only have a very small number of independent points per unit distance in the vertical relative to the number in the horizontal domain.

Figure 1 illustrates this scale discrepancy using the example of a large-amplitude atmospheric gravity wave (GW) observed over the Southern Ocean in June 2010. Figure 1a shows the GW with the vertical axis stretched, as it would typically be shown
in scientific publications and related contexts. Figure 1b shows the same GW with its true aspect ratio, demonstrating that the complex vertical structure visible in Figure 1a is entirely contained within an extremely narrow vertical layer close to the Earth's surface. As a result of these very different characteristic scales, it is technically challenging to study vertically-varying atmospheric structures, such as GWs, using remote-sensing techniques.

This anisotropy causes major problems when using standard spectral analysis techniques, such as the Fourier Transform
and the S-Transform (Stockwell et al., 1996), to study the vertical variations associated with GWs in these data. The GWs observable by these instruments also have vertical scales greater than 10 km and horizontal scales of tens to hundreds of kilometres, but the relatively small number of measured points in the vertical domain relative to the horizontal, combined with the short vertical window range available in the data as imposed by the actual depth of the stratosphere and mesosphere, leads to severe limits on estimates of GW vertical wavelengths $\lambda_z$ when spectral analyses of this type are applied. An example of
this issue is seen in Figure 4e of Wright et al. (2017), where a 3D S-Transform method is applied to AIRS data, but shown to





only be capable of resolving two unique values of $\lambda_z$ in a volume where we would expect significant local $\lambda_z$ variations based on geophysical and measurement-science grounds.

In this study, we address this problem by developing, implementing and testing a new spectral analysis method based on phase differences between two-dimensional S-Transforms of individual horizontal data levels. Our new method is able to

characterise the vertical wavelengths of GWs measured in 3D nadir-sensed satellite data to a much higher degree of precision, with no significant reduction of data quality for any other analysed variables such as horizontal wavelength and wave amplitude. Although we only apply it here to the problem of GWs in nadir-sensed data, the technique is fully applicable to any spectral analysis problem involving multi-dimensional datasets where one of the dimensions is significantly lower-resolution than the others.

In Section 2 we describe the data we use for this study, and in Section 3 outline our method, both conceptually and as we have implemented it computationally. Section 4 then tests our method on simulated data, after which Section 5 applies it to three observational case studies. Finally, we summarise and discuss our results and draw conclusions in Section 6.

## 2   Data

We primarily use real temperature observations from the Atmospheric InfraRed Sounder (AIRS) on NASA's Aqua satellite,

retrieved using the hgh-resolution retrieval method described by Hoffmann and Alexander (2009). These data have been extensively used for previous GW studies (e.g. Ern et al., 2017; Wright et al., 2017; Jackson et al., 2018; Hindley et al., 2019). We also (Section 4) analyse simulated waves sampled using the measurement geometry of AIRS, for which the wave properties are by definition fully known in advance and which therefore provide a robust quantitative test.

AIRS is a nadir-sensing instrument, and is thus well-optimised geometrically to observe fine horizontal structures. In the

horizontal, the scan track has a width of 1780 km, varying in resolution from $\sim$13.5 km $\times$ 13.5 km at nadir to 41 km $\times$ 21.4 km at track edge. The resolution and track width are imposed by the scanning geometry, orbital altitude and hardware design.

In the vertical, the Hoffmann and Alexander (2009) retrieval uses the 15 $\mu$m and 4.3 $\mu$m infrared $CO_2$ channels to derive a vertical temperature profile for each measurement footprint. This retrieval is optimised for stratospheric GW studies, with noise and resolution balanced for a vertical resolution between 7 km – 20 km over the 15 km – 60 km altitude range. The retrieval

switches between a daytime mode, which uses the 15 $\mu$m channel only, and a nightmode mode which uses both channels; this switch is due to the assumption of local thermodynamic equilibrium required for the underlying scattering calculations being violated during daytime due to solar excitation of $CO_2$ molecules. This switch has an effect on our analysis, manifesting as horizontal 'stripes' of reduced magnitude seen in the input data at the 33 km and 42 km levels in Figures 7 and 8 which propagate through to our final output products. Figure 2 of Hindley et al. (2019) illustrates the vertical resolution and retrieval

noise as a function of height for a range of regimes.

As such, GWs are often clearly-resolved in horizontal cuts through the AIRS measurement volume, but this is usually not the case in the vertical. While an observationally-resolved wave may vary over many tens of individual points in the horizontal domain, the entire useful vertical extent of the data is only about 15 data points, many of which overlap in information content.



These issues strongly inhibit the use of 3D spectral analysis tools such as the S-Transform, and as such make the dataset an
ideal application for our new approach.

## 3  Methods

The main conceptual tool underpinning this study is the use of phase shifts between spectral features to compute wavelengths.
This concept was previously applied to satellite studies of gravity waves by Alexander and Barnet (2007) and Alexander
et al. (2008), who used phase shifts between one-dimensional S-Transforms to compute wavelengths in two-dimensional nadir
sounder data and one-dimensional limb-sounder data respectively. Wright and Gille (2013) expanded upon this work to measure
multiple overlapping waves in the same 1D input data signal, and although we do not do so here the same method could be
applied to our new method to determine properties of multiple spectral features at each point in the 3D data volume.

Our new method, which we refer to throughout this study as the "2D+1 ST" for brevity, builds instead on the 2D S-Transform
(2D ST) of Hindley et al. (2016), and also uses concepts from the 3D S-Transform (3D ST) of Wright et al. (2017) as modified
by Hindley et al. (2019). The "+1" here refers to the use of the phase shift for computations made in the vertical dimension.
The 2D ST and 3D ST are both multi-dimensional modifications to the 1D S-Transform of Stockwell et al. (1996), which allow
us in turn to extend the phase-shift approach to higher dimensionalities - although we do not do so here, the same conceptual
approach could, for example, be used to identify temporal periods between 3D volumes of observed or modelled data. Note
that, in common with these previous studies, we use an exhaustive approach to analysing the S-Transform spectral space rather
than a discrete orthonormal approach which, while computationally faster, might miss subtleties of the near-resolution-limit
structures we intend to analyse.

### 3.1  Data Preprocessing

AIRS data are not uniform in geographic space. In the horizontal domain, this is due to measurement geometry - the measure-
ments are uniform in instrument-angular space, but distorted in geographic space by the curvature of the Earth - and in the
vertical it is due to the relatively-lower vertical resolution of the Hoffmann and Alexander (2009) retrieval at the highest and
lowest altitudes in the lowermost stratosphere and lower mesosphere

Since most spectral analyses, including the ST, require regularly-gridded data, we must therefore preprocess the AIRS data
to make this the case. We do this by first removing altitude levels below 20 km and above 60 km - leaving regularly-vertically-
spaced data with a 3 km level-to-level spacing, since all of the non-regularly-spaced levels are outside this range - and then by
regridding each level independently onto a regular spatial horizontal grid with the same number of elements (135 along-track
by 90 across-track pixels) as the input data but with a uniform inter-point spacing in each direction.





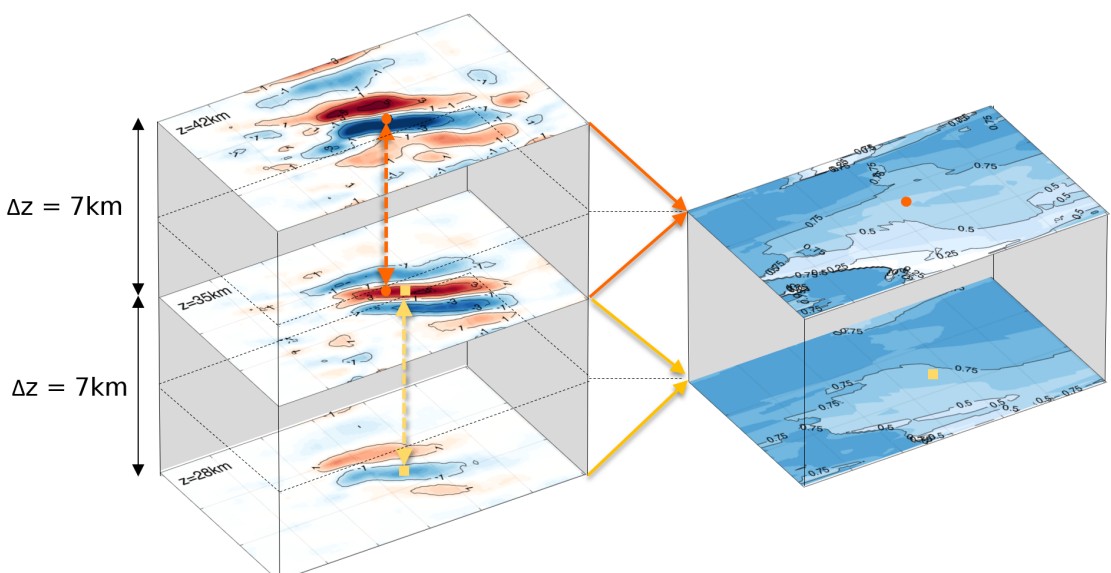

**Figure 2.** Diagram illustrating the underlying concept of our method. Left column show three horizontal cuts through a large-vertical-scale gravity wave, with a vertical spacing between levels of 7 km. Values shown at each level are the magnitude of a gravity wave signal, with red indicating a positive magnitude and blue a negative magnitude. All units except for height are arbitrary. Right column shows cospectral phase differences between the wave structures at each height, on the same horizontal domain; these are derived from a 2D S-Transform analysis, with contours showing phase difference in units of $\pi$ radians. Yellow and orange arrows and markers indicate examples of how $\lambda_z$ is computed in our analysis, and are explained in the main text.

### 3.2 2D+1 ST Concept

Figure 2 illustrates the underlying concept of the 2D+1 ST. Shown on the left of the figure are three separate height levels measured through a large-vertical-scale gravity wave, with each level separated by a vertical distance of 7 km. Horizontal
distance and amplitude units are arbitrary.

For each of these three height levels, we compute the 2D ST $S$ (not shown). This returns a 4D object, where the first two dimensions represent physical distances in the original domain $(x,y)$ and the latter two dimensions represent wavenumber combinations $(k_x, k_y)$. For each pair of adjacent levels, we then find the complex cospectrum

$$C_{i,i+1} = S_i S_{i+1}^{\dagger}, \tag{1}$$

where $S_i$ and $S_{i+1}$ are the complex 2D STs of layer $i$ and $i+1$, and the $^{\dagger}$ indicates complex-conjugation. Note that, while in this example the levels used to calculate the cospectrum are vertically adjacent, this is not a requirement, and in our analyses below we use a different vertical step size.

From this cospectrum we then compute a covarying amplitude $A = \sqrt{C_{i,i+1}}$ and a phase difference

$$\Delta\phi = \frac{Im(C_{i,i+1})}{Re(C_{i,i+1})}, \tag{2}$$





where $Im$ and $Re$ denote taking the imaginary and real component respectively. The resulting $A$ and $\Delta\phi$ are both four-dimensional objects, which we collapse to two dimensions by identifying the largest amplitude signal in $(k_x, k_y)$ for each $(x, y)$. The resulting two-dimensional $A$ and $\Delta\phi$ thus represent, respectively, the amplitude of the strongest covarying wavelike signal at each pixel in the spatial domain between the two levels and the phase difference between these level-pixels.

The right column of Figure 2 shows the phase differences $\Delta\phi$ between the two level-pairs, with the orange (yellow) solid
arrows showing the two height levels contributing to the upper (lower) $\Delta\phi$ map. Phase differences shown are absolute (i.e. unsigned), and shown in units of $\pi$ radians, i.e. a value of 0.5 indicates a phase difference of $\pi/2$ radians. The choice to make $\Delta\phi$ uniformly positive here is to avoid introducing mathematical signs into our example calculations, and for most practical GW applications it would instead be sensible to force $\Delta\phi$ to be negative, implying upward wave propagation (Wright et al., 2016, 2017; Ern et al., 2017).

From these phase differences, we can then compute $\lambda_z$ between the two levels for each pixel $(x, y)$. Two examples of these are shown, indicated by the yellow squares/dashed arrows and by the orange circles/dashed arrows. We consider first the orange example. The pixels highlighted by the orange circle are in a negative node of the wave in the 42 km layer and in a positive node in the 35 km layer. For this pair of pixels, the phase difference (orange circle on the right upper layer) is $0.55\pi$. This implies $\lambda_z$ for the wave spanning this pixel-pair of $(2/0.55 \times 7)$ 25 km. Following a similar logic for the yellow example, we
obtain $\lambda_z$ between the lower pair of levels of 21 km. Equivalent estimates of $\lambda_z$ can be computed for every pixel on every pair of levels in the data.

### 3.3 2D+1 ST Calculation Procedure

The above example demonstrates the concept of the 2D+1 ST, but is not suitable for general application to data without refinement. In particular (a) use of a standard 2D ST will cause wave amplitude to be significantly underestimated (Wright
et al., 2017; Hindley et al., 2016, 2019), (b) when applied to AIRS data retrieved using the method of Hoffmann and Alexander (2009) it is unwise to use adjacent vertical levels in the data for the calculation as they are not sufficiently independent, and (c) if we calculate the 2D ST for all possible horizontal wavenumbers $(k_x, k_y)$ at each vertical level then the total runtime of the procedure can become extremely large.

Therefore, for these reasons and to incorporate the computational improvements made to the 3D ST analysis by Hindley
et al. (2019), we use the following procedure to compute our 2D+1 ST outputs for AIRS.

1. We first identify the dominant 1000 $(k_x, k_y)$ combinations in the 3D volume of the data, as described by Hindley et al. (2019). This dramatically reduces the number of independent 2D wavenumber combinations we subsequently need to analyse by removing combinations for which the signal content is very low, which in turn very significantly reduces total runtime with no noticable loss of output quality.

2. Next, we independently compute the 2D ST $S$ of each vertical level in the data for these 1000 dominant wavenumber combinations.





3. Following this, we iterate over height levels, computing and storing the complex cospectrum $C$ between the vertical level above and the vertical level below, i.e. $C_{i-1,i+1}$. This two-vertical-level step increases the independence of the two input levels to the calculation relative to a one-vertical-level step, improving output accuracy. Nominally, this choice prevents us from measuring short $\lambda_z$, but as the vertical level spacing in Hoffmann and Alexander (2009) AIRS data is much smaller than the actual vertical resolution of the instrument, no meaningful information is lost.

4. The resulting cospectra are somewhat noisy, due to the high levels of pixel-level noise in the AIRS data relative to typical gravity wave amplitudes. Accordingly, we next average each cospectrum (in 4D) with the cospectra for the levels vertically above and below, using a Gaussian weighting in height which we scale using an analogue to the 3D ST weighting parameter $c$ (Hindley et al., 2016; Wright et al., 2017). In our software implementation this parameter is scaled such that a value of 1 corresponds to a FWHM of a single 3 km level, 0.5 indicates 2 levels, etc. This value is fully adjustable, and can be used to tune the analysis in the presence of high relative noise.

5. Finally, we identify the dominant covarying signal between each pair of vertical layers $i-1$ and $i+1$, by selecting the wavenumber combinations $(k_x, k_y)$ with the largest absolute spectral amplitude $A$ in the cospectrum $C_{i-1,i+1}$, as described in Section 3.2. As in Section 3.2, we can then compute phase difference $\Delta\phi$, $\lambda_z$ and wave amplitude $A$ for each pixel on each level. We also obtain the horizontal wavenumbers $k_x$ and $k_y$ from the spectral locations at which the maxima in $A$ were computed. This step incorporates the amplitude-boosting approach of Hindley et al. (2019), which corrects for the amplitude reduction of a standard ST.

In both the simulated and observed analyses below, we use a vertical weighting value of 0.25, corresponding to a FWHM of 12 km (i.e. four input data levels). The 2D ST analyses at each level also use a tunable level of $c$ for each horizontal dimension, and in common with previous work we also set these values to 0.25.

## 4 Simulated Data

We first assess our new technique against a set of simulated waves. This is intended to help us identify any systematic biases or limitations to the technique before moving on to study observed cases.

### 4.1 Creating the Artificial Wave Field

Figure 3 illustrates the method we use to produce our test waves.

We first create an empty volume with the same dimensions as a real AIRS granule, by loading a real AIRS granule and setting all temperature values to zero. Before setting the temperatures to zero, the granule is interpolated to a regular spatial grid, in the same way as the true AIRS measurements discussed below.

We next create an artificial sinusoid to fill this volume, with a specified temperature amplitude $T'$, three-dimensional wavelength $(\lambda_x, \lambda_y, \lambda_z)$ and rotation $(\theta_x, \theta_y, \theta_z)$ around a point at the centre of the volume in that dimension (defined as positive in the anticlockwise direction). We iterate through a wide range of each of these values in our analyses below.

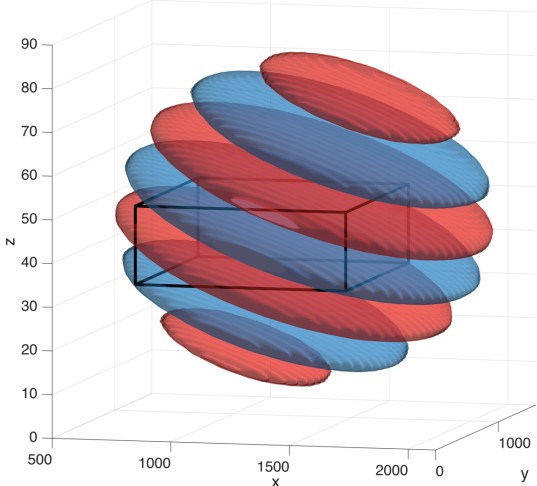

**Figure 3.** Example figure illustrating the method used for our artificial wave tests. Each wave is a 3D sinusoid with a specified amplitude, three-dimensional wavelength $(\lambda_x, \lambda_y, \lambda_z)$ and rotation $(\theta_x, \theta_y, \theta_z)$ relative to the axes of a 3D Cartesian grid of the same volume as a typical AIRS granule. Red and blue surfaces show 3D contours of fixed positive (red) and negative (blue) magnitude; values are arbitrary but show the same absolute values for red and blue. Each output parameter is averaged across the region at the centre of the volume indicated by the black box, with dimensions $800 \times 800 \times 18$ km.

The sinusoid is then weighted with a Gaussian of maximum amplitude 1 and full-width-at-half-maximum $1200 \times 950 \times 50$ km centred at the centre of the granule. This is intended to ameliorate any wraparound effects due to the use of Fast Fourier Transform algorithms in the 2DST and 3DST.

We then separately apply the 3DST and 2D+1 ST to the artificial wave. We do this for both the fully-artificial field and also a 'realistic noise' case; this case is produced by adding the detrended $T'$ field of a real granule (granule 001 of the 8th of May 2008) to the fully-artificial wave, in order to simulate a realistic combined detrending and random noise pattern. This basis granule was empirically selected as one that exhibited no visibly-apparent coherent structures that either the 2D+1 ST or 3DST was likely to identify as a GW. In previous work (Wright, 2010) we have demonstrated that S-Transform analyses are highly robust to non-systematic noise, and thus we expect differences between using granule or another as our basis to be very small.

Finally, to assess the accuracy of the ST outputs, we take the median value of each output field over a $800 \times 800 \times 18$ km box at the centre of the wave (shown as a black box in Figure 3). The median is chosen to better characterise the different effects of the 3DST and 2D+1 ST on estimated $\lambda_z$ in our results (Figure 4l, discussed below), and the subregion is chosen to avoid both wraparound effects and regions where our Gaussian taper has reduced the wave signal to below the noise floor in the realistic-noise cases.

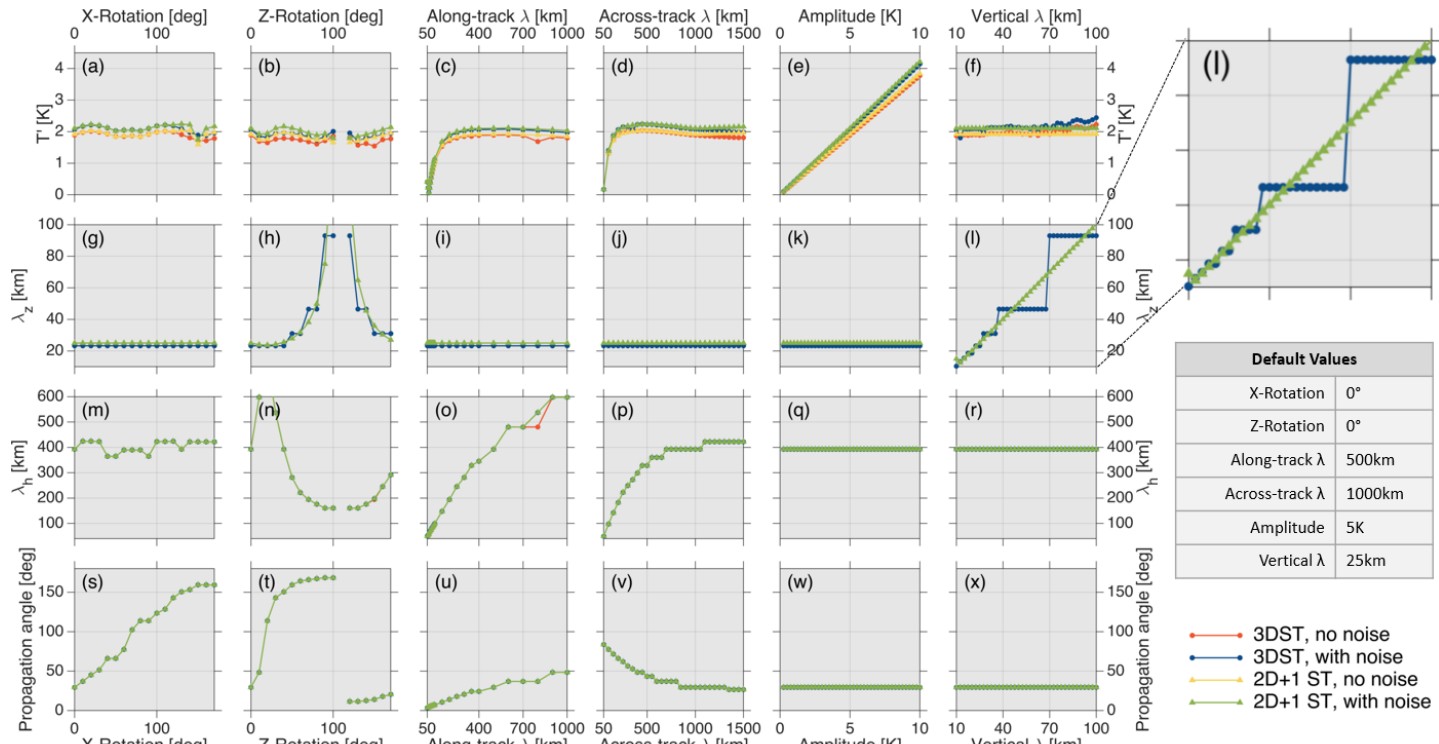

**Figure 4.** Systematic comparison of the 3DST and 2D+1 ST for a wide range of input artificial waves. Each panel shows recovered (a-f) wave amplitude (g-l) $\lambda_z$ (m-r) $\lambda_h$ (s-x) propagation angle, for the four different analyses indicated at bottom right. For each panel, the parameter named at the top and bottom of the column is varied systematically, with all other parameters set to the default values shown in the table at top right. Note that, in many cases the results for each method are identical, and for these only the last-plotted (2D+1 ST with noise) analysis is visible. (l) is the only test to exhibit significantly different results between the 3DST and 2D+1 ST, and is shown enlarged at top right.

.

## 4.2 Test Results

Using the above approach, we estimate wave amplitude $T'$, $\lambda_z$, $\lambda_h$ and wave propagation angle $\theta$ for a battery of artificial waves. The results of these analyses are shown in Figure 4 for four methods: the 3DST (orange), the 2D+1 ST (yellow), the
195 3DST with realistic noise (blue) and the 2D+1 ST with realistic noise (green).

We have specifically tested the effects on these output variables of variations in input $T'$, $\lambda_z$, along-track and across-track wavelengths $\lambda_x$ and $\lambda_y$, and rotations about the centre of the granule in the $x$, $y$ and $z$ directions. We omit rotation in $y$ from Figure 4 as these results simply reproduce features of rotations in $x$. In all cases, we have varied one of these parameters systematically while holding all others constant at the values shown in the table shown at top right of Figure 4; these default
200 values were chosen as representing values typical of waves seen in previous AIRS GW studies. Tests with other values of these





default parameters (not shown) gave similar relationships between the input and output variables. Values below the spatial resolution of AIRS (specifically, horizontal wavelengths of less than 50 km and vertical wavelengths below 10 km) are omitted.

In general, the results for all four methods are extremely close, and in many cases, particular for output $\lambda_h$ and output propagation angle, they are identical. Rotations in $z$ give the largest effects on all output variables with the exception of output amplitude (Figure 4h,n,t); this is because $z$-rotations significantly change the relationship between $\lambda_z$ and $\lambda_h$ of the artificial wave, with consistent results across all four methods.

In amplitude (Figure 4a-f), the 2D+1 ST measures very slightly higher amplitudes than the 3DST in both the no-noise and noise cases, suggesting that the 2D+1 ST may be recovering very slightly more of the input wave amplitude than the 3DST. However, this difference is small ($\sim$5% at most, and usually less). In all cases, the measured amplitude is significantly lower than that of the input wave (typically by a factor of around 2.5); this is not entirely due to S-Transform amplitude reduction, but is also related to the Gaussian windowing we apply across our range (which will reduce amplitudes at all points away from volume-centre) and the choice to generate the wave on the AIRS measurement grid (which will cause us to undersample the theoretical peak magnitudes of the artificial waveform).

The largest differences between tests in the analysis output are seen in output $\lambda_z$ when we systematically vary input $\lambda_z$ (Figure 4l). In the 3DST analyses (only the with-noise version is visible as the lines exactly overlap), $\lambda_z$ is well-recovered at input $\lambda_z < 30$ km, with output $\lambda_z$ directly corresponding to this input. Above this value, however, output $\lambda_z$ begins to stall at certain wavelengths as input $\lambda_z$ increases, with the measured values remaining at fixed output $\lambda_z$ values over increasingly long ranges of input $\lambda_z$. In contrast, output $\lambda_z$ for the 2D+1 ST increase smoothly through this range, with no noticable stepping.

This difference is due to the underlying approach used in the two methods. The 3DST uses Fourier Transform algorithms in the vertical direction, which are inherently limited to resolving integer modes of the data window width. For example, if we were to have 40 points in the vertical, then the 3DST would only be able to measure wavelengths of 40/1, 40/2, 40/3, etc. The phase-difference approach used in the 2D+1 ST, by contrast, measures instead the difference in horizontal phase between each selected pair of levels. It is thus able to measure a continuous range of $\lambda_z$, as we see in Figure 4l.

As a tradeoff, the 2D+1 ST cannot measure very short vertical waves where the phase change $\Delta\phi$ between the two levels is greater than $\pi$ radians. For AIRS data, such as we are simulating here, this is not a consideration: the vertical resolution of the instrument is coarser than the wavelength at which this problem arises unless inter-level step sizes of greater than two are used. However, it must be carefully considered if applying this technique to other datasets or with a different step size. In such cases (not shown), measured vertical wavelengths will be longer than in the underlying data, due to aliasing of phase differences into the $\pm\pi$ range.

For these artificial waves, we therefore conclude that the 2D+1 ST as implemented here performs equivalently to the 3DST, with the exception of $\lambda_z$ where the 2D+1 ST performs better provided (not shown) the magnitude of the phase change between the levels we take the cospectrum of is less than $\pi$ radians. This condition is usually true for a two-level step in AIRS data.





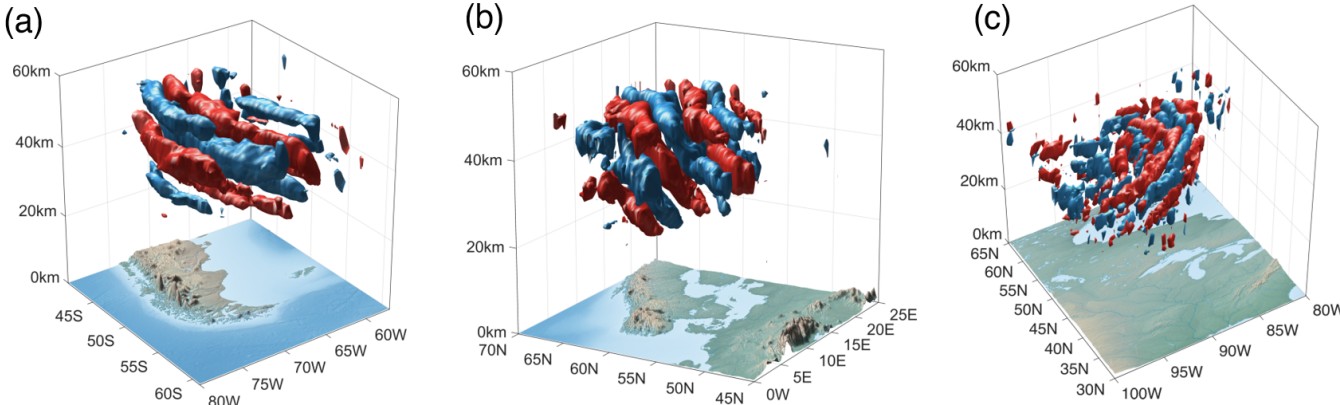

**Figure 5.** 3D sketches of the three waves used as case studies; amplitudes have been omitted as these are shown in cross-section below (Figures 6–8). Viewing angles have been selected to best highlight the 3D phase structure of each case.

## 5 Case Studies

We next demonstrate our method on three case study waves, specifically (a) a large mountain wave over the southern part of South America, (b) a large mountain wave over Scandinavia and (c) a convective wave over the US and Canadian Midwest. These three examples have been selected to represent a range of geophysical challenges, in order to provide a broad test of the 2D+1 ST approach as compared to the 3DST.

### 5.1 Large Andean Mountain Wave

Figures 5a and 6 illustrate our first example, a large orographic wave over South America observed at approximately 06:00 UTC on the 6th May 2008. This wave has been used as a baseline test of 3D GW analysis methods in previous studies (Wright et al., 2016, 2017), and thus represents a well-understood case is already contextualised within the scientific literature.

The wave is strongly and clearly visible across the whole AIRS altitude range (Figure 6a,h), with vertically-slanted phase fronts aligned to indicate a source on the west coast of South America. It has a very large amplitude, with magnitudes peaking at $> \pm 15\,\mathrm{K}$ (note that the colour scale saturates at $\pm 10\,\mathrm{K}$), and is angled across the AIRS scan track but with a significant along-track component. It is thus highly suited to study using AIRS data.

For all non-input fields, we have set voxels (i.e. 3D pixels) with amplitudes $<2.7\,\mathrm{K}$ data after smoothing with a $5\times5$ voxel horizontal-domain-only boxcar to zero, in order to remove noise-dominated regions and focus on resolved wave structure. The boxcar size and amplitude cutoff have been selected empirically, but can also be considered on physical grounds: the smoothing ensures extremely-localised variations do not produce distracting gaps in plotted regions, while the 2.7 K value is approximately twice the AIRS retrieval noise floor at the central height in our data range, and thus this approach will retain packet-edge locations where the smoothing would average these values to below the real noise floor. Most large-scale studies



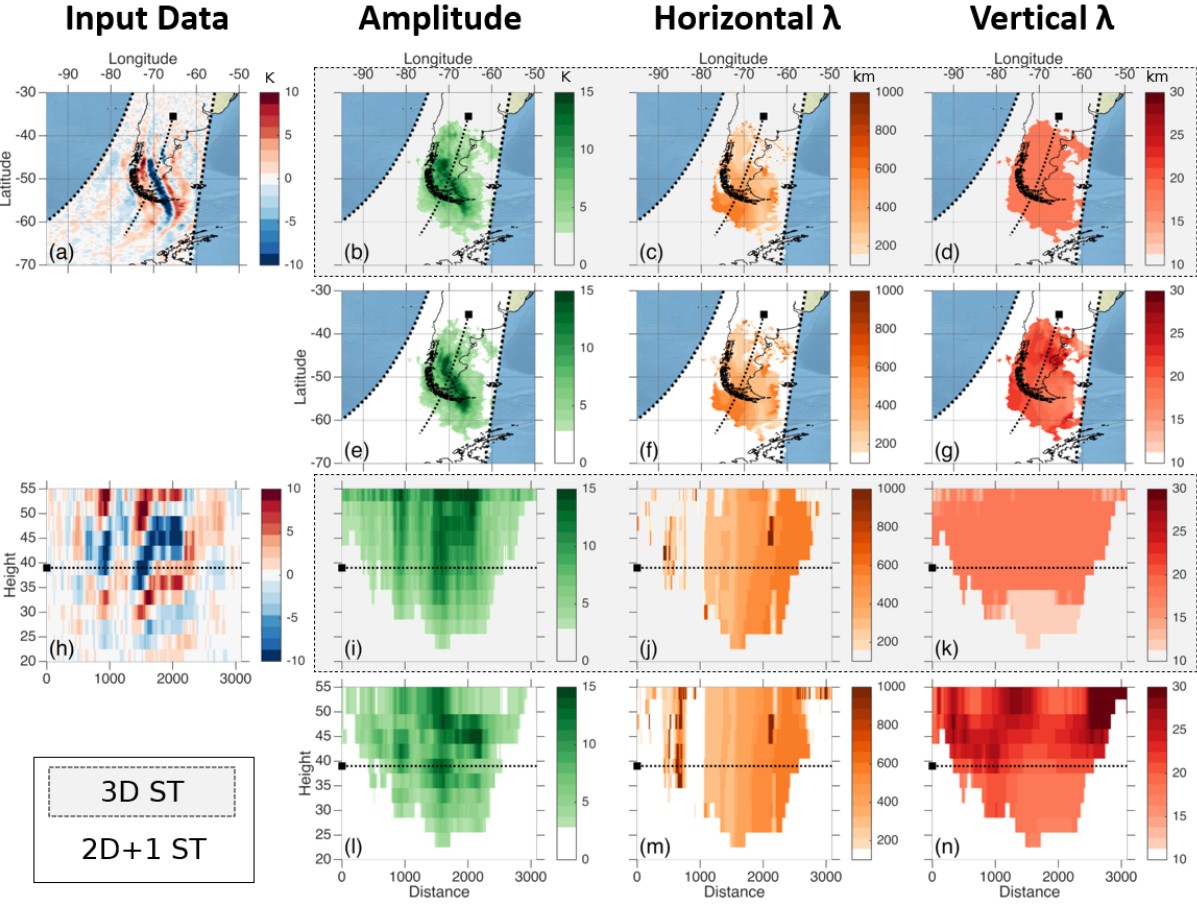

**Figure 6.** Comparison of 3D ST and 2D+1 ST estimates of GW parameters for a large orographic gravity wave observed in May 2008 over South America. (a) and (h) show original data, as a map at 39 km altitude and a vertical cut along the instrument track respectively; dotted lines show the location of the other set of panels, with a black square indicating zero-distance on the lower panel. (b-d,i-k) 3DST estimates, (e-g,l-n) 2D+1 ST estimates of (b,e,i,l) wave amplitude (c,f,j,m) $\lambda_h$ (d,g,k,n) $\lambda_z$. Units for each panel are shown above the colourbar at the top of each row. Data have been boxcar-smoothed by 3 voxels in each direction in the horizontal plane before plotting to reduce visible noise. Regions with smoothed output amplitudes below 2.7 K have been removed from all panels except (a) and (h) to focus on wave features only.

of GWs in AIRS data (e.g Ern et al., 2017; Hindley et al., 2020; Perrett et al., 2021) impose similar minimum-amplitude criteria before including a given voxel in their analyses.

In 3DST output (Figure 6), the wave is clearly visible, with a well-defined region of high wave amplitude resolved over

Chile, Argentina and the surrounding seas. We also see associated patches of well-defined $\lambda_h$. The same wave was studied using an earlier version of our 3DST implementation by Wright et al. (2017), and only small differences are visible between their old and our revised 3DST outputs, primarily higher measured amplitudes.



2D+1 ST $\lambda_h$ is very similar to that obtained from the 3DST, with the exception of a region of long $\lambda_h$ at low-altitude in the northern part of the along-track cut (Figure 6m). This region both exhibits very low wave amplitude relative to the rest of the

wave shown and is only sporadically filled from voxel to voxel in both analyses; thus, both analyses are likely largely fitting to background noise rather than true wave signals in this region despite our output filtering.

Larger differences in output are seen for amplitude and $\lambda_z$. Amplitude is significantly more spatially-localised for the 2D+1 ST than for the 3DST, likely arising from the calculation of amplitude independently for each level-pair, and this is discussed further in the general case below.

$\lambda_z$, Figure 6(d,g,k,n), exhibits the largest difference, consistent with our intentions behind developing the 2D+1 ST approach. 3DST estimates are limited to Fourier modes of the vertical window, which in practice allow only two unique values across the whole domain in both the horizontal and vertical slices, specifically 20 km and 13 km. The measured $\lambda_z$ for every voxel is one of these two values. This effect was also visible in Wright et al. (2017). The 2D+1 ST output, on the other hand, exhibits a much broader spectrum of $\lambda_z$, with values distributed across a continuum. Smooth variations in $\lambda_z$ can be clearly seen in both

the mapped (Figure 6g) and sliced (Figure 6n) fields, as would be expected in the real atmosphere.

As examples of features in the horizontal plane which can be identified in the 2D+1 ST output but not the 3DST output, we clearly resolve a region of longer $\lambda_z$ over the Golfo San Jorge (67°W, 45°S) and a region of shorter $\lambda_z$ over Tierra del Fuego (69°W, 55°S). In the vertical plane, we see a smooth increase in $\lambda_z$ across the height range, suggesting lengthening of $\lambda_z$ with height. This effect can be discerned from the 3DST output as a discontinuity between a low-$\lambda_z$ region below 30 km altitude and

a longer-$\lambda_z$ region above this, but the rate of growth can be clearly quantified in the 2D+1 ST version. We can also determine that growth occurs gradually across the entire range rather than discontinuously across a narrower height range around 30 km, which was not possible with 3DST output alone. Even in central altitude regions ∼39 km altitude, where AIRS data have their best resolution and lowest noise, the 2D+1 shows more realistic variations in $\lambda_z$ with horizontal distance.

For this case study, therefore, we conclude that $\lambda_h$ data quality is largely unchanged and that $\lambda_z$ data quality is significantly

improved. Amplitude differs between the two versions, but our comparison does not provide sufficient information to identify if this change is an improvement or a reduction in output quality.

## 5.2   Scandinavian Mountain Wave

Our second example, Figures 5b and 7, is an orographic wave over Scandinavia observed shortly after 12:00 UTC on the 13th of January 2007. This wave is also clearly visible in AIRS data, but has a much smaller amplitude than our first example and

also covers a smaller geographic area, thus providing a more exacting test.

As with the Andean case above, the wave is made up of clearly-defined parallel phase fronts, sloping upwards at an angle to the AIRS scan track. Phase-front magnitudes in the raw AIRS data are larger at low altitude, with a slight reduction in amplitude above 40 km altitude. Amplitude measurements from both the 3DST and 2D+1 ST are consistent with this, with maxima at around 35 km altitude. These maxima are tightly spatially localised around the southern half of Norway and western

parts of central Sweden.



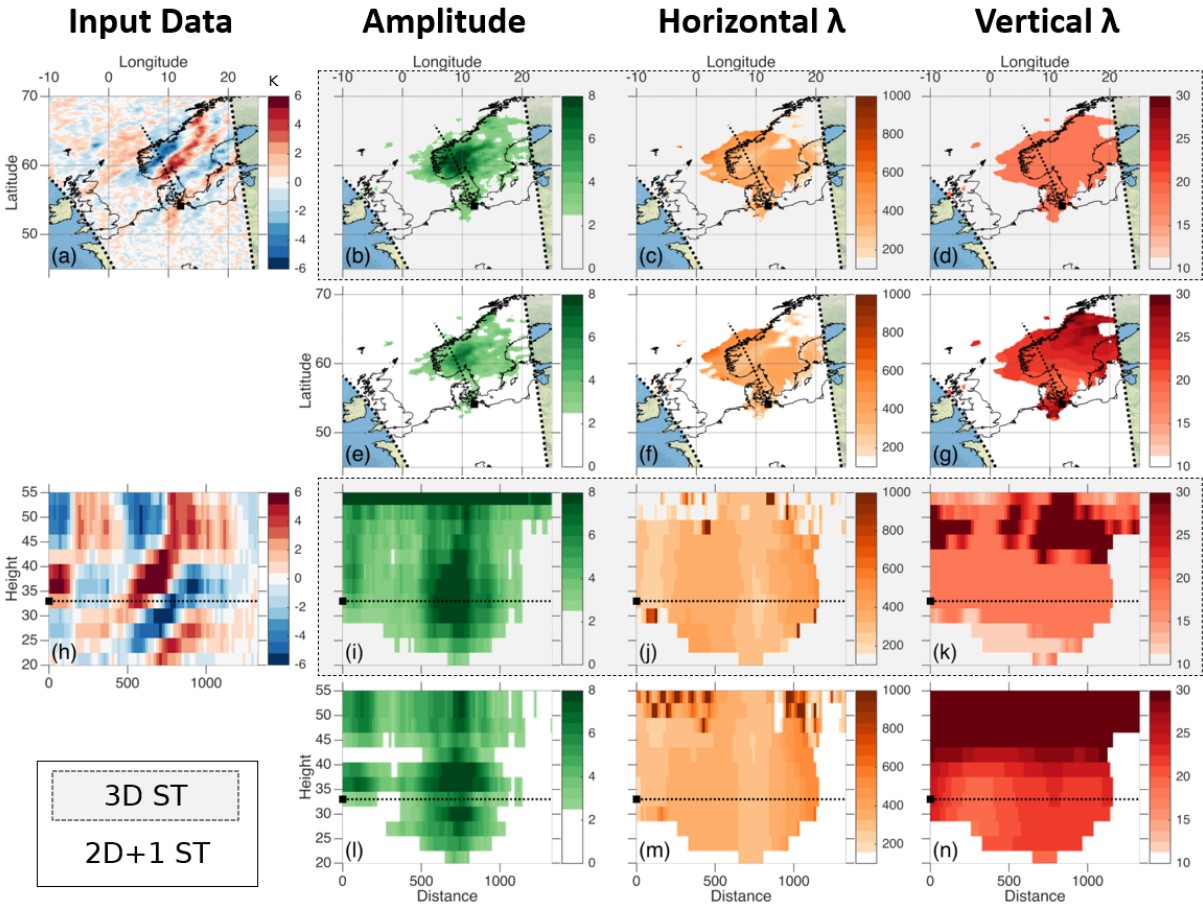

**Figure 7.** As Figure 6, but for an orographic wave observed over Scandinavia in January 2007 and with horizontal cuts taken at 33 km altitude. Note that in panel (n) $\lambda_z$ at high altitudes very significantly saturates the colour scale; this is discussed further in the text.

Both analyses show minor flaws in their output amplitude field, with the 3DST producing a patch of extremely large amplitudes at the top of the plotted altitude range and the 2D+1 ST exhibiting two horizontal stripes, at ~33 km and ~42 km altitude. In the latter case, these stripes are caused by similar horizontal stripes visible in the input data at these altitudes (Figure 7h), which the 3DST is able to compensate for by implicitly using information from surrounding height levels, but which the 2D+1 ST is unable to compensate for as individual level-pairs are used. We therefore argue that the 2D+1 ST output is more faithful to the input data, while the 3DST output is likely a more accurate description of the true atmospheric feature in this individual case. The ultimate origin of these stripes is due to the use of the daytime rather than nighttime retrieval, as discussed in Section 2.

Both analyses see the same along-track pattern of $\lambda_h$, with longer $\lambda_h$ between 200-600 km along-track from our origin (black square), then shorter $\lambda_h$ for around 200 km before increasing again until the edge of the wave packet is reached. As with



the previous example, the 2D+1 ST has a tendency to produce longer estimates of $\lambda_h$ in regions of low-amplitude, i.e. without significant wave activity - in this case this is most visible at the top-left of Figure 7m, where the input temperature variations have become near-vertical and amplitudes are at their lowest above-cutoff values.

3DST $\lambda_z$ for this example exhibits three unique values, increasing with height except for a small layer at the top of the data. The 2D+1 ST output again exhibits a continuum of values, increasing steadily across the height range. At heights below 40 km, 2D+1 ST output $\lambda_z$ across the whole height range is slightly higher than the 3DST, but this difference is consistent with values obtained from visual examination of Figure 7h and thus reflects a better estimate of the true wave properties.

Estimated values of $\lambda_z$ increase rapidly at the top of the altitude range, significantly saturating the colour scale to reach maximal values above 50 km altitude of $\sim$120 km at along-track distances between 500-100 km and >150 km elsewhere. These values are extremely large, but consistent with the input data (Figure 7h), where the phase fronts are near-vertical at these altitudes. This may be a real effect, or it may be an artefact of the significantly reduced instrument vertical resolution at these altitudes (see e.g. Figure 2 of Hindley et al. (2019)). However, the same effect is not seen in our other case studies and furthermore is seen here as a continuous extension of the phase fronts present in the lower part of the stratosphere (note that our data are detrended independently at each height level), suggesting at least some contribution from the real atmosphere.

Finally, as with the previous example, with the 2D+1 ST we can measure structural variations in the $\lambda_z$ field which were invisible to the 3DST. In this case, a previously-invisible feature is a reduction in $\lambda_z$ between the main Scandinavian ridgeline ($\sim$17°E, 67°N) and the lower-lying regions of Sweden heading towards Våstergötland and the Øresund ($\sim$10°E, 56°N), which while nominally visible as a step-change in the 3DST output follows a clear gradient in the 2D+1 ST output. Indeed, a north-south gradient in $\lambda_z$ is clearly seen across the whole of southern Sweden in the 2D+1 ST output, which is reduced to a single switch of mode in the 3DST.

### 5.3 North American Convective Wave

Our final example, Figures 5c and 8, is a convection-generated wave observed between 8:00 and 9:00 UTC over North America on the 8th of July 2008. This wave is a much more challenging test of our method than the previous two case studies for four reasons:

1. the wave has a maximum amplitude of $\sim$5 K, which is small compared to the previous examples and thus closer to the instrument noise floor;

2. the wave is curved about an arc rather than well-aligned with the instrument frame of reference, and hence changes rapidly in along-track and across-track wavelength in the instrument frame of reference;

3. the wave is close to the edge of the AIRS swath, in a region where across-track horizontal resolution is around a factor of two below that at track-centre; and

4. the wave is close to the day-night terminator, where the Hoffmann and Alexander (2009) retrieval switches mode, and thus measurements of this wave are potentially complicated by a discontinuity in noise level and spatial resolution only



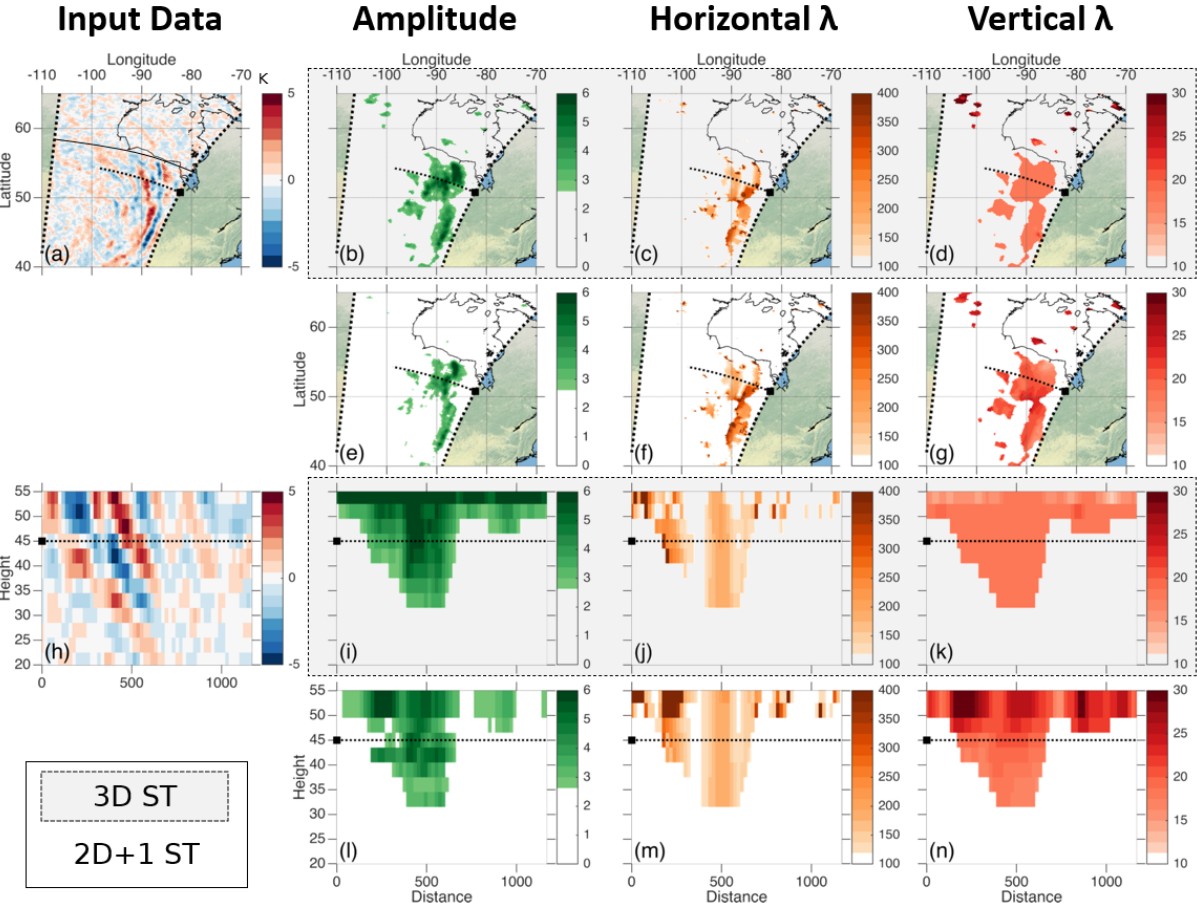

**Figure 8.** As Figure 6, but for a convective wave observed over North American in July 2008 and with horizontal cuts taken at 45 km altitude. A discontinuity is visible in some panels between a region on the equatorward side of the maps and a region on the poleward side: this is due to a switch between the nighttime and daytime retrievals, which have very different noise characteristics.

a few hundred kilometres away from the wave centre. This manifests itself in Figure 8a as a reduction in the input temperature range poleward of a cross-track line approximately along the southern edge of Hudson Bay, indicated by a

black solid line.

The wave is visible in Figure 8a as a series of arcs on the eastern side of the AIRS swath, with radii centred at a point somewhere between Calgary and Medicine Hat, both in Alberta - specifically, the curvature of the arcs suggests the origin to be around 112°W, 51°N, slightly off the left side of the map. Convective cloud data from AIRS' 8.1 $\mu$m channel (Supplementary Figure S1) show evidence of convective activity in this region throughout this day and the day before, and analysis of the

relative 2D horizontal wavelengths in the along-track and cross-track directions (not shown) further confirm that the wave is likely to be radiating from around this location. Our maps are shown at the 45 km altitude level, but at lower-stratospheric



altitudes the wave is challenging to distinguish from background without prior knowledge of its location, and is at the very noise limit of AIRS data.

Consistent with this difficulty, amplitudes in both analyses are largest at altitudes above 35–40 km. Above-background

amplitudes are also tightly spatially-localised around the narrow observed wavefronts. There is perhaps some visual evidence in Figure 8a that the wave arc may continue at much lower amplitudes poleward of the terminator, but this is not visible at above-cutoff levels in the output amplitude data; visual analysis of the full below-noise amplitude data fields (not shown) also do not show evidence that these features are detected as waves by either analysis method.

$\lambda_h$ is near-identical for the two analyses, with the maps showing a region of $\lambda_h \sim$200–350 km along the wave front and the

slices through the wave showing a tight region of $\lambda_h \sim$200 km at all heights around the magnitude peak. This band extends throughout the entire height range from 20–60 km; since each level-pair in the 2D+1 ST analysis is treated independently, this demonstrates that, while amplitude and discernability from background of the wave are poor at low altitudes, ST-based analyses are still capable of measuring $\lambda_h$ even against a high relative noise floor.

Finally, and consistently with all previous tests, we see $\lambda_z$ steadily increasing across the height range, with a unique measured

value $\lambda_z$ for the 3DST over almsot the whole data extent and a continuum for the 2D+1 ST.

## 6   Summary, Discussion and Conclusions

In this study, we have described and implemented a new spectral analysis method (the "2D+1 ST") for the characterisation of wavelike signals in three-dimensional data. This method is particularly well-suited to data which is fine in two dimensions but coarse in the third, which is very often the case for satellite measurements of the Earth's atmosphere, and to data where

the feature size to be extracted is large relative to the total domain length. This method uses a two-dimensional Stockwell Transform (Stockwell et al., 1996; Hindley et al., 2016) in the well-resolved two-dimensional plane, with wavelengths in the coarse third dimension computed from phase differences between S-Transformed two-dimensional levels.

We have tested this new method on artificial waves, both with and without noise, and on observed case-study waves in real data from NASA's AIRS instrument. In all cases, the 2D+1 ST is almost identically capable to the 3D ST for determining fine-

dimension (usually horizontal) wave structure, and much better suited to characterising coarse-dimension (usually vertical) structure. This improvement is primarily because the phase-difference approach used is not restricted to Fourier modes of the domain size.

2D+1 ST wave amplitudes are equivalent in magnitude to those obtained from the 3DST, and more responsive to local variations in input wave amplitude. This is in principle superior to the 3DST, as the inputs are better reflected in the outputs.

However, in real cases it may represent a slight degradation in the direct usefulness of results, as the broader contextual information used in the 3DST can help to ameliorate localised data quality deficiencies in the input field. This is however a small effect.

Although we do not discuss it in the body of the article, the 2D+1 ST is also computationally slower than the 3DST, due both to the need to (1) analyse each level with an individual 2DST and (2) to take phase differences between levels, with the





latter having a larger overall impact on runtime. This may affect the choice of which algorithm to use when considering large

volumes of data. For the three case studies considered, the 2D+1 ST typically takes twice the runtime of the equivalent 3DST

on the two-core system used for this study; as the S-Transform parallelises well, this relative increase may be smaller on nodes

with more available cores. The runtime difference could be further ameliorated by reducing the number of individual frequency

combinations considered, at a small cost to output accuracy.

We conclude, therefore, that the 2D+1 ST in general outperforms the 3DST for measuring waves in data with at least one

coarse dimension, a category which includes almost all current-generation Earth observation satellite data. In particular, the

2D+1 ST has two major advantages over the 3SDST for application to satellite observations such as AIRS and similar nadir

sounders, viz:

1.  It allows us to identify spatial variations in measured $\lambda_z$ that may have physical significance for the geophysics of the

wave environment. In our case studies, we have been able to identify regional changes in $\lambda_z$ that were not visible or

only weakly visible to the 3DST, e.g. finding longer wavelengths over the Golfo San Jorge, and shorter wavelengths over

Tierra del Fuego (Figure 6g), and a general north-south reduction in wavelength over the west of Sweden) (Figure 7g),

in addition to generally much better characterisation of $\lambda_z$ growth with height.

2.  It is much more suited to the measurement of extremely long vertical wavelengths, including those that are longer than

the visible length of the wave in the measured data. This capability is demonstrated both for test waves (Figure 4l) and

practically at high altitudes in our Scandinavian case study. This is of significant benefit for measuring portions of the

gravity wave spectrum that are extremely hard to observe, including for example hypothesised extremely-long-$\lambda_z$ waves

which may act to transport large quantities of surface momentum directly to the upper atmosphere.

In future work, we intend to apply this new technique to other satellite and ground-based datasets, with the dual aims of

better characterising internal variability in the stratospheric and mesospheric gravity wave spectrum, and of identifying and

quantifying the effects of long-$\lambda_z$ waves on upper-atmospheric dynamics.

*Code and data availability.* All code produced for and data used in this study, with the exception of the topography and surface-imagery

datasets used to contextualise Figures 1 and 5–8, have been archived with Zenodo and can be accessed via doi:10.5281/zenodo.4569067

*Author contributions.* Alexander provided the original suggestion for the work. Wright produced the figures, wrote the initial manuscript

draft, and developed an initial algorithmic approach in collaboration with Alexander and Holt. Hindley heavily revised the algorithmic

approach to be more computationally efficient and significantly more accurate. Hoffman produced and supplied the AIRS data used in our

examples. All authors contributed to methodological development and to the final manuscript.





*Competing interests.* The authors have no competing interests.

*Acknowledgements.* Funding to support this work was received from the following research grants: Royal Society University Research
Fellowship UF160545 (supporting Wright), NERC grants NE/R001391/1 and NE/S00985X/1 (supporting Wright and Hindley). We also
acknowledge vital underpinning discussions at and within the context of "New Quantitative Constraints on Orographic Gravity Wave Stress
and Drag", an International Team project supported by the International Space Science Institute, and during an NWRA-funded visit to
Boulder by Wright to collaborate on developing an initial approach with Alexander and Holt.



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
