# Peer review of "Using Vertical Phase Differences to Better Resolve 3D Gravity Wave Structure"

_Atmospheric Measurement Techniques, 2021_

## Author Comment (AC1)

**Response to Reviewers**

We thank the reviewers for their comments and the time spent preparing them.

**Reviewer 1**

**General comments**

***1. The task is challenging, as very few independent measurement points in the vertical are available, and the new methods main advantage is for long wavelengths. However, the GW spectrum of vertical wavelengths extends well below 10 km, and the limitation of the instrument to wavelengths above 10 km cannot be overcome by data analysis. It could however be acknowledged that this limitation still holds, and that other remote-sensing techniques exist that indeed yield much higher vertical resolution (the statement in p. 2, l. 32).***

The word 'satellite' has been added to narrow the context of the statement, as (e.g.) lidar is indeed much better at measuring small vertical features.

***2. Your focus is entirely on gravity waves, with a tendency to "extremely vertically long waves" (p. 1, l. 13). Can you comment on the discrimination of other types of waves like planetary waves and tides in AIRS data, are these of no relevance, or is there danger of confusion? On the same note, you state that GW with 70 km vertical wavelength transport significant moentum to the upper atmosphere (p. 1, l. 18),but don't add a citation. In lidar analysis, GW vertical wavelenghts are often only analyzed below 20 km vertical wavelength (to avoid named confusion with tides and planetary waves) and the majority of gravity waves observed has 5-15 km vertical wavelengths.***

Tides and planetary waves should have removed using the detrending method discussed below. In principle the method could be used to measure the vertical structure of tides and planetary waves in the raw (i.e. un-detrended) data, or in other datasets, although we do not do so here. In practice, tides are also expected to be very small at the altitudes studied.

We have added two citations to demonstrate deep wave momentum transfer; this has also been added to p18 l392 in response to Reviewer Two's similar comment.

***3. When you sketched your algorithm (p. 6, l. 117), I wondered if the presence of monochromatic waves is a requirement. Would it give meaningful results in more complex environments, like a superposition of waves with different wavelengths, or is it required that it is indeed one and the same wave contained in both levels? And is this the usual case in AIRS data, or do you usually see more complex patterns than the examples selected?***

The presence of monochromatic waves is not a strict requirement, but while detecting superposed waves is technically possibly with this approach, it would in practice causes a very large increase in computational load. The reason for this is that these waves would have distinct spectral peaks; therefore, the simple approach of taking the maximum in the 2DST fields at each level could no longer be used and instead a full peak finding algorithm would be required to operate within the 2D spectral space created by each voxel. We have demonstrated that this works in 1D (Wright and Gille 2013, doi:10.1002/grl.50378), but doing so in 2D is significantly more computationally complex and therefore we have not (yet!) implemented it computationally. Assuming such a peak-finding approach within the spectral plane associated with each voxel were implemented, the 2D+1 method should however be entirely able to measure the phase difference between levels of each wave separately. as the full

4D co-ordinates of each peak would be known - essentially, the distinctive elements of this approach relative to the 3DST would not be expected to degrade the result at all.

***4. Have you considered comparing the three cases to reanalysis data, e.g. ERA5? Your method incorporates smoothing in the vertical, and I wonder if there is substantial variation in the vertical that is not resolved by AIRS and your method, and what it means for the interpretation of your results.***

This is an interesting question, and Wright and Hindley are currently supervising a PhD student who plans to do exactly this for the more general case beyond these three examples. However, her work commenced after submission of the manuscript and is still at a very early stage. Therefore, in order to allow timely publication and (perhaps more importantly) to keep the manuscript focused on the method itself rather than a detailed description of differences between AIRS and ERA5, we propose that this work should be kept separate for a later article.

***5. You stress the applicability of the method to other types of datasets with similar scale relations - besides the atmosphere, can you think of an example? (p. 3, l. 49)***

A hypothetical example might be a pair or small set of ADCPs measuring ocean-internal waves across a range of depths - here, the fine dimensions would be time and depth, and the coarse dimension the horizontal separation between the ADCPs.

***6. In p. 5, Fig 2 you said the level-to-level spacing is 3 km, but you show 7 km-space levels in the figure, which is not a multiple?***

As this diagram is conceptual rather than intended specifically to illustrate the AIRS case, this is not *a priori* a problem. However, clearly it is a source of some unnecessary confusion, so it makes sense to change it to numerically correspond to the specific case of AIRS. Accordingly, the numbers on the plot and in the associated text have been changed to correspond to a 6 km spacing, i.e. two AIRS levels of separation. The data shown have not been changed, as the plot is intended to be a conceptual example only rather than illustrating a specific case, and changing the data as well may introduce unintended errors elsewhere in the text.

***7. In p. 5, l. 210 you mean that there is, although you have corrected the retrieved amplitude (as stated in l. 163), still a factor 2.5 is missing? You gave a number of reasons, but it still seems large?***

The largest cause of the difference is due to the way the wave has been produced rather than the analysis method: the nominal amplitude of the wave as a pure sine wave is significantly reduced when the wave is computed onto a coarse discrete grid. After allowing for this effect, the remaining errors due to the method are very small.

***8. Maybe you can reformulate the sentence in p. 4, l. 90, e.g. what do you mean by "discrete orthonormal approach". The sentence just wasn't very clear to me.***

The sentence has been rewritten, and two example references added.

**Technical corrections**

***A. p. 1, l. 1 "Atmospheric gravity waves... are a dynamical process" - I am not a native speaker, is it correct to say that a wave is a process? I would rather say that i.e. the transport of momentum or the breaking of a wave is a process.***

To avoid any ambiguity this has been changed this to 'mechanism'.

***B. p. 2, Fig. 1 please add the information that perturbations in temperature are shown (I know, it is obvious, and it says "K", but still, for the readers that are not too familiar***

*with gravity waves)*

Fixed.

*C. p. 3, l. 55, "hgh" -> "high"*

Fixed.

*D. p. 5, Fig. 2 "The left column shows three..."*

Fixed.

*E. p. 7, l. 150 please give numbers for the step (2 x 3 = 6 km?, or 2 x 7 = 14 km?), for the minimum vertical wavelength resolved, and the actual vertical resolution*

See response 6, above.

*F. p. 8, l. 186 "using one granule or another"?*

Fixed.

*G. p. 8, l. 187 "the median value" do you mean the median value of retrieved vertical wavelength?*

In this case the comment is general to all properties - e.g. where the panel shows amplitude it would be amplitude, where the panel shows horizontal wavelength it would be horizontal wavelength, etc.

*H. p. 9, Fig. 4 add commas and "and" in "wave amplitude T', (g-l) vertical wavelength $\lambda_z$, (m-r)..., and (s-x)...", same in p. 12, Fig. 6*

Fixed.

*I. p. 11 l. 241 "case that is"?*

Fixed.

*J. p. 11, l. 246 please check this sentence again, what are "non-input fields", can't you omit "data" in "< 2.7 K data"*

"Non-input fields" has been changed to "fields other than the input temperature perturbations" to remove any confusion, and the extraneous 'data' has been removed.

*K. p. 12, Fig. 6 caption: "box-car smoothed by 3 voxels": in the text it was 5x5 voxels?*

Fixed (now $5 \times 5$).

*L. p. 17, l. 355, "almost"*

Fixed.

*M. p. 18, l. 382 "3SDST" -> "3DST"*

Fixed.

**Reviewer 2**

**General comments**

*1. It is apparent from your discussion and figures, that the input to your method are temperature fluctuation fields not temperature field. However, neither the Data section (section 2) nor the Data Preprocessing section (section 3.1) state how these temperature fluctuations are derived from the AIRS temperature retrieval output. Please add a section on the detrending for instance in either of the before mentioned sections.*

The detrending method used is a fourth-order cross-track polynomial filter, as used in a wide range of previous AIRS studies. As this has been used extensively before, it is probably not

necessary to add a detailed description, but it should definitely be mentioned! So, a clause has been added to Section 2 saying this and referencing the previous studies that have used it for more details.

**Technical comments**

***A. p.3,l.55: hgh-resolution -> high-resolution***

Fixed.

***B. p.4,l.80: You mention here, that the method could be expanded to consider multiple overlapping waves in the signal. How would your current method deal with an overlapping wave signal? Can you elaborate on possible effects that a "less" monochromatic signal would have on your results?***

See response 3 to Reviewer One.

***C. p.6,l.136: I was wondering why it is "unwise to use adjacent vertical levels". What is the dependence between the levels? Could you clarify?***

***D. p.7,l.150: You state that your choice of vertical level selection is limiting the vertical wavelength you retrieve. What is the minimum vertical wavelength that you can retrieve? And how does this affect the wave spectra you are retrieving?***

In both cases, this is a choice made based on the nature of the AIRS retrieval, rather than due to the method presented. AIRS can resolve waves with vertical wavelength $\sim$8 km or coarser, but the data are stored on 3 km levels. Hence, the information content of (for example) odd-numbered levels is not independent from the adjacent even-numbered levels, and taking a phase difference between them would yield unphysical information. The practical effect of this is to limit the minimum resolvable wavelength to 6 km, but in practice the data cannot contain such features anyway.

***E. p.7,l.156: The abbreviation FWHM is used here before definition.***

Now defined at first mention.

***F. p.11,l.235: Maybe refer to the same wording in the headlines following as chosen here (or the other way around) for consistency.***

Done.

***G. p.11,l.241: . . . case is already . . . -> case, which is already ?***

I think either is fine, and prefer the original to keep the pace of the sentence (the added comma breaks it up a little for my taste). I'm very happy to change this though if the reviewer has a strong preference the other way.

***H. caption of Fig. 6: "(a) and (h) show original data". . . I assume these are temperature perturbations? Can you specify that?***

Specified.

***I. "Data have been boxcar-smoothed by 3 voxels" -> The text states 5 voxel smoothing? Please make this consistent.***

Fixed (to 5x5).

***J. p.15,l.309: 500-100km -> 500-1000km***

Fixed.

***K. p.18,l.382: 3SDST -> 3DST***

Fixed.

***L. p.18,l.392: You are mentioning a hypothesis of „extremely-long" vertical wave-***

***length here. Could you add a reference?***

See Response 2 to Reviewer One.

***M. p.18,l.387: There is a loose ) after Sweden***

Fixed**)**.

---

## Author Response (AR2)

**Response to Reviewers**

We thank reviewer 1 for their additional comments, which we respond to individually below.

1. ***I suggest to add a comment on the minimum resolvable wavelength. Both reviewers asked to clarify this, and (although it is obvious) it is worthwile confirming that the method can do nothing to overcome this limitation.***

The text "but with a lower limit (i.e. minimum resolvable wavelength) of twice the level spacing." has been added to the paragraph starting on line 225 of the revised version.

2. ***I suggest to re-add the paragraph on the computation of the vertical wavelength. The readers might figure it out themselves, but it was a helpful description nonetheless, and it is advertised in the figure caption. I don't know why it was removed.***

The removal of this paragraph was a typographical error by the lead author, which was not noticed until a few days after resubmission by which point the article would already have been sent out to reviewers. It has been restored in this draft to the original unchanged version.